# Influencing Factors on the Quality of Lymph Node Dissection for Stage IA Non-Small Cell Lung Cancer: A Retrospective Nationwide Cohort Study

**DOI:** 10.3390/cancers16020346

**Published:** 2024-01-13

**Authors:** Piotr Gabryel, Piotr Skrzypczak, Magdalena Roszak, Alessio Campisi, Dominika Zielińska, Maciej Bryl, Katarzyna Stencel, Cezary Piwkowski

**Affiliations:** 1Department of Thoracic Surgery, Poznan University of Medical Sciences, Szamarzewskiego 62 Street, 60-569 Poznan, Poland; 2Department of Computer Science and Statistics, Poznan University of Medical Sciences, Rokietnicka 7 Street, 60-806 Poznan, Poland; 3Department of Thoracic Surgery, University and Hospital Trust–Ospedale Borgo Trento, Piazzale Aristide Stefani 1, 37126 Verona, Italy

**Keywords:** lung cancer, early-stage, surgery, thoracoscopy, VATS, minimally invasive surgery, lobectomy, lymphadenectomy, mediastinal lymph node dissection, quality

## Abstract

**Simple Summary:**

The excision of the lymph nodes, known as lymphadenectomy, is an essential part of surgical operation for lung cancer. This nationwide study was conducted to identify factors influencing the quality of lymphadenectomy. We included 4271 patients who underwent a minimally invasive surgical operation for early-stage lung cancer. Our findings indicate that the requirements for lymphadenectomy were satisfied in 27.9% of patients. Statistical analysis revealed that patients who underwent positron emission tomography–computed tomography before surgery, patients with larger tumors, and those operated on by experienced surgeons had a higher accuracy of lymphadenectomy. Additionally, the study revealed a declining trend in the quality of lymphadenectomy over time. Importantly, more extensive lymph node excision did not correlate with elevated complication rates or in-hospital mortality. In light of these findings, it is imperative to implement actions aimed at enhancing the quality of lymphadenectomy for lung cancer.

**Abstract:**

Lymphadenectomy is an essential part of complete surgical operation for non-small cell lung cancer (NSCLC). This retrospective, multicenter cohort study aimed to identify factors that influence the lymphadenectomy quality. Data were obtained from the Polish Lung Cancer Study Group Database. The primary endpoint was lobe-specific mediastinal lymph node dissection (L-SMLND). The study included 4271 patients who underwent VATS lobectomy for stage IA NSCLC, operated between 2007 and 2022. L-SMLND was performed in 1190 patients (27.9%). The remaining 3081 patients (72.1%) did not meet the L-SMLND criteria. Multivariate logistic regression analysis showed that patients with PET-CT (OR 3.238, 95% CI: 2.315 to 4.529; *p* < 0.001), with larger tumors (pT1a vs. pT1b vs. pT1c) (OR 1.292; 95% CI: 1.009 to 1.653; *p* = 0.042), and those operated on by experienced surgeons (OR 1.959, 95% CI: 1.432 to 2.679; *p* < 0.001) had a higher probability of undergoing L-SMLND. The quality of lymphadenectomy decreased over time (OR 0.647, 95% CI: 0.474 to 0.884; *p* = 0.006). An analysis of propensity-matched groups showed that more extensive lymph node dissection was not related to in-hospital mortality, complication rates, and hospitalization duration. Actions are needed to improve the quality of lymphadenectomy for NSCLC.

## 1. Introduction

According to the World Health Organization data, lung cancer is currently one of the most common malignancies and the leading cause of cancer-related death worldwide [1]. It is typically diagnosed at an advanced stage and, in most cases, surgical treatment is not feasible [2]. Consequently, substantial efforts have been made in recent years to improve lung cancer detection and treatment. The introduction of low-dose chest computed tomography screening has increased the detection rate of early-stage lung cancer, enabling more frequent qualification for surgical treatment and reducing lung cancer mortality [3,4]. Video-assisted thoracoscopic surgery (VATS) has improved both the early- and long-term outcomes of surgical operations, compared to open approaches [5,6,7]. Robotic-assisted surgery may yield even better outcomes, including more accurate lymphadenectomy and lower conversion rates compared to VATS [8,9], but the high cost of the procedure may impede its widespread implementation [10]. Concerning the type of anatomical resection, lobectomy has been considered the standard of care for early-stage non-small cell lung cancer (NSCLC) since the study by Ginsberg et al. [11]. Recent trials have demonstrated comparable outcomes of segmentectomy for peripheral IA1 and IA2 NSCLC [12,13]. However, segmentectomy is a complex procedure suitable only for small, intrasegmental tumors, and may be associated with increased local recurrence rate [14,15]. Due to the limitations of robotic surgery and segmentectomy, VATS lobectomy will likely continue to be the procedure of choice for the majority of the early-stage NSCLC patients worldwide.

In addition to lung anatomical resection, mediastinal lymph node dissection (MLND) is essential for achieving complete NSCLC resection. The International Association for the Study of Lung Cancer (IASLC) guidelines recommend one of two types of MLND, either systematic (SMLND) or lobe-specific (L-SMLND) [16]. However, there is no consensus on whether one type of MLND is superior to the other [17]. While SMLND is believed to be more accurate in detecting mediastinal lymph node metastases [18], Deng et al. demonstrated that stage cIA NSCLC most often metastasize to the lymph nodes specific for a given lobe of the lung [19]. Shapiro et. al. found that in patients with cN0/N1 NSCLC, pN2 metastases followed the lobe-specific pattern and the recurrence rate was similar after L-SMLND and SMLND [20]. Moreover, recent meta-analyses demonstrated that L-SMLND, compared to SMLND, can be associated with a lower incidence of postoperative complications, such as chylothorax and arrhythmia, comparable recurrence-free survival and favorable overall survival [21,22]. These findings support the role of L-SMLND for the treatment of selected patients with early-stage NSCLC.

Despite the evidence, intraoperative nodal assessment very often falls short of recommendations [23]. In a study by Ray et al., only 18% of patients operated on for early-stage NSCLC had any type of MLND, and 6% had nodal sampling. The remaining 75% of patients had an intraoperative nodal assessment that met neither standard. Moreover, the authors found that MLND was associated with superior survival compared to less comprehensive nodal staging [24]. These findings were supported by the recent study that demonstrated that among patients after VATS lobectomy for stage IA NSCLC, only 28.8% met the criteria for L-SMLND. More importantly, the study also found that L-SMLND was related to higher 5-year survival rates compared to patients with less comprehensive nodal assessment [25].

The reasons for poor-quality lymph node assessment for early-stage NSCLC are not well understood. Some studies found that demographic factors, comorbidities, cancer stage, type of surgery, and the type of surgical equipment may influence MLND [26,27,28]. These studies, however, included heterogenous populations of patients operated on at all stages of lung cancer, with both minimally invasive and open approaches. Factors that could influence the quality of MLND during VATS lobectomy for early-stage NSCLC are poorly researched. As the quality of MLND significantly affects the long-term outcomes of lung cancer treatment, and the number of minimally invasive procedures for early-stage lung cancer is likely to increase, research in this area may be of great clinical importance. The objectives of this study were to evaluate the quality of MLND in patients undergoing VATS lobectomy for early-stage lung cancer, identify factors related to the quality of MLND, and to evaluate the impact of MLND on postoperative complications.

## 2. Materials and Methods

The Bioethics Committee of the Poznan University of Medical Sciences waived the need to obtain consent for the collection and analysis of the anonymized data, and for the publication of the results of this multicenter, retrospective, cohort study.

### 2.1. Diagnosis, Surgical Treatment, and Reporting of Treatment Results for Lung Cancer in Poland

Lung cancer in Poland is diagnosed by pulmonologists and thoracic surgeons, as part of outpatient, as well as in-hospital specialist medical care. The public healthcare system provides good access to lung cancer treatment. All decisions regarding oncological treatment, including non-surgical and surgical treatment, are made by multidisciplinary tumor boards. In Poland, approximately 4000 patients are operated on for lung cancer annually, constituting about 20% of all newly diagnosed cases of the disease [29]. Operations are carried out in 29 thoracic surgery centers covering the country’s total population of 38 million people. Anatomical lung resection can only be performed by board-certified thoracic surgeons, or residents under the supervision of a thoracic surgeon. Data regarding surgical treatment are obligatorily entered by residents and consultants into the Polish Lung Cancer Study Group Database. The database includes data on patients’ demographics, comorbidities, pulmonary function tests, radiological tests, invasive staging, surgical approach, type of operation, number of lymph nodes and lymph node stations removed, results of histopathological examination, postoperative complications, chest tube duration and postoperative hospital stay, neoadjuvant and adjuvant treatment, and follow-up. TNM stage is automatically calculated based on the entered data, and updated to the most recent version, in this case to the eighth edition, of the TNM classification.

### 2.2. Study Design

This retrospective, multicenter cohort study included patients who underwent VATS lobectomy for pathological stage IA lung cancer between 17 April 2007 and 14 October 2022. Data for the study were obtained from the Polish Lung Cancer Study Group Database. The study excluded patients after induction treatment (chemo-, immuno-, and/or radiotherapy), with an open approach, after sublobar resection, extended resection or pneumonectomy, with histology other than NSCLC (benign histology, metastases, or small-cell lung cancer), and with missing information on the surgical approach, dissected lymph nodes, histological type, or postoperative care.

For the purpose of the study, L-SMLND was defined according to the IASLC guidelines, and included the following nodal stations for each of type of lobectomy:Right upper paratracheal (2R), right lower paratracheal (4R), and subcarinal (7) nodes for right upper and right middle lobectomy, and for right upper bilobectomy.Right lower paratracheal (4R), subcarinal (7), and paraesophageal (8) or pulmonary ligament (9) nodes for right lower lobectomy.Aorto-pulmonary window (5), paraaortic (6), and subcarinal (7) nodes for left upper lobectomy.Subcarinal (7), paraesophageal (8), and pulmonary ligament (9) nodes for left lower lobectomy.All right-sided stations (2R, 4R, 7, and 8 or 9) for right lower bilobectomy.

Systematic mediastinal nodal dissection included stations 2R, 4R, 7, 8, and 9 for the right-sided lobectomy and stations 5, 6, 7, 8, and 9 for the left-sided lobectomy.

The study cohort was divided into two groups. Patients were assigned to either the L-SMLND group, or to the non-L-SMLND group, depending on whether the IASLC criteria for lobe-specific mediastinal nodal dissection were met.

The primary endpoint of the study was the lobe-specific mediastinal lymph node dissection (L-SMLND). The study analyzed the relationship between preoperative and surgery-related variables, and the accuracy of the lymph node assessment. The secondary endpoints were postoperative complications and other variables related to the postoperative course.

### 2.3. Statistical Analyses

The analyzed data were expressed as mean ± standard deviation, median, minimum, maximum values, interquartile range (Q1 lower quartile, Q3 upper quartile), or percentage, as appropriate. The relationship between variables was analyzed using Spearman’s rank correlation coefficient. The normality of distribution was tested using the Shapiro–Wilk test and equality of variances was checked using Levene’s test. A comparison of two unpaired groups was performed using the unpaired t-test for data that followed a normal distribution and had homogeneity of variances or the Mann–Whitney U-test. Categorical data were analyzed using the χ2 test when the sample size was larger than 40 and all expected values were greater than ten; for other situations, the exact test of Fisher or χ2 test with Yate’s correction was used. All results were considered significant at *p* < 0.05. Factors that obtained *p* < 0.05 in univariate analysis were then analyzed in logistic regression. Differences in perioperative outcomes between types of lymph node assessment were evaluated using a propensity-matched analysis of non-L-SMLND versus L-SMLND. Propensity scores were developed, defined as the probability of treatment assignment, conditional on age, sex, Charlson Comorbidity Index, Thoracic Revised Cardiac Risk Index, chronic obstructive, pulmonary disease, surgeon’s experience, and type of lobectomy. All covariates were determined to be clinically relevant. Using a nearest neighbor matching algorithm without replacement with a caliper of 0.01, the most appropriately matched pairs were selected. Data manipulation and all calculations were performed in IBM^®^ SPSS^®^ Statistics version 27th (PS Imago Pro 8).

## 3. Results

The study comprised 4271 patients who underwent VATS lobectomy for stage IA non-small cell lung cancer and met all study criteria. The rates of postoperative complications and in-hospital mortality were 21.9% and 0.54%, respectively. The median postoperative chest tube duration was 3 days (IQR 2 to 4 days), and the hospital stay duration was 6 days (IQR 5 to 7 days).

Lobe-specific mediastinal lymph node dissection (L-SMLND) guidelines were met in 1190 patients (27.9%). The remaining 3081 patients (72.1%) did not meet L-SMLND criteria. Patients with L-SMLND had higher numbers of lymph nodes removed (median 12 vs. 8, *p* < 0.001) and lymph nodes stations dissected (median 6 vs. 4, *p* < 0.001).

Univariate analysis revealed that the L-SMLND criteria were more likely to be met in patients with a smoking history (*p* < 0.001), a higher body mass index (*p* = 0.04), chronic obstructive pulmonary disease (*p* < 0.001), and larger diameter tumors (T1a vs. T1b vs. T1c; *p* < 0.001). The quality of nodal assessment was also influenced by preoperative staging procedures—patients with preoperative positron emission tomography–computed tomography (PET-CT, *p* < 0.001) were more likely to meet L-SMLND criteria, while patients who had undergone endobronchial ultrasound-transbronchial needle aspiration (EBUS-TBNA, *p* = 0.038) were less likely. Mediastinoscopy was associated with a higher L-SMLND rate (*p* = 0.023), but the number of subjects with this procedure was low and the results should be interpreted with caution. L-SMLND was more frequently performed in patients undergoing right-sided lobectomy (*p* < 0.001) by surgeons with greater surgical experience, i.e., who performed more than 50 VATS lobectomies (*p* = 0.003). Interestingly, patients operated on more recently exhibited lower quality lymphadenectomy and were less likely to meet L-SMLND criteria. This relationship was evident both when the study period was chronologically divided into four periods of three years each (*p* < 0.001; Figure 1a) and when the study group was chronologically divided into four groups of equal sizes (*p* < 0.001; Figure 1b). The decreasing quality of lymphadenectomy was associated with the increasing number of surgeons performing VATS lobectomy (Figure 1c,d). The baseline, surgical, postoperative, and histopathological characteristics, along with the results of univariate analyses, are presented in Table 1 and Table 2.

The variables that demonstrated significance in the multivariate logistic regression analysis were PET-CT (OR, 3.238; 95% CI: 2.315 to 4.529; *p* < 0.001), pT (OR, 1.292; 95% CI: 1.009 to 1.653; *p* = 0.042), surgeon’s experience (OR, 1.959; 95% CI: 1.432 to 2.679; *p* < 0.001), and period of surgery (OR, 0.647; 95% CI: 0.474 to 0.884; *p* = 0.006). Patients who underwent PET-CT, had larger tumors (pT1a vs. pT1b vs. pT1c), and were operated on by experienced thoracic surgeons exhibited a higher probability of meeting L-SMLND criteria. Conversely, the likelihood of meeting L-SMLND criteria decreased in patients operated on more recently. The results of the multivariate analysis are presented in Table 3.

Lastly, we assessed the relationship between the extent of lymph node dissection and postoperative course. To minimize bias, propensity score matching was performed, resulting in two groups with similar basic characteristics, each comprising 1184 patients (Appendix A). Analysis of propensity-matched groups showed that L-SMLND did not significantly influence the postoperative course and the complication rates. The chest tube duration was slightly longer in the non-L-SMLND group compared to the L-SMLND group (4 days, IQR 3 to 5 days vs. 3 days, IQR 2 to 5 days; *p* < 0.001). Other variables, including hospital stay duration, in-hospital death, and complication rates did not differ between the groups (Table 4).

## 4. Discussion

Several studies have demonstrated that in many patients undergoing surgery for lung cancer, intraoperative lymph nodes assessment does not meet generally accepted guidelines. This study aimed to identify factors related to the quality of MLND in patients undergoing surgery for early-stage NSCLC. Analyzing all VATS lobectomies for stage IA NSCLC performed in one large European country from 2007 to 2022, we found that the L-SMLND criteria were met in 27.9% of patients. Patients with PET-CT, larger tumors (pT1a vs. pT1b vs. pT1c), and those operated on by experienced thoracic surgeons, were more likely to meet L-SMLND criteria. However, we observed an unfavorable trend of decreasing lymphadenectomy quality over the years. Furthermore, the study revealed that patients with L-SMLND had a higher numbers of mediastinal lymph nodes and lymph node stations removed. L-SMLND was not associated with postoperative complications rates, in-hospital mortality, or hospitalization duration.

The current guidelines advocate for mediastinal lymph node dissection for lung cancer surgery. This study has substantiated that various factors influence the MLND accuracy. Familiarity with these factors provides an opportunity to adapt surgical procedures in order to improve the quality of treatment.

PET-CT holds a pivotal role in clinical staging and is endorsed by multiple lung cancer diagnostic guidelines [30,31]. PET-CT facilitates identification of lymph nodes suspected of cancer metastases, enabling precise targeting of invasive pre- and intraoperative lymph node assessment [32]. Our study demonstrated higher accuracy of lymphadenectomy among patients who underwent PET-CT. These findings underscore the importance of PET-CT in the diagnosis and treatment of lung cancer. PET-CT should be performed in all patients before anatomical resection for lung cancer, unless there are absolute contraindications.

This study also found that L-SMLND rates were lower for smaller lung nodules. These findings align with those of Pawelczyk et al. and Edwards et al., demonstrating that the adequacy of intraoperative nodal staging for NSCLC was lower in patients with T1a tumors [33,34]. Small nodules seldom metastasize to lymph nodes, likely explaining why surgeons are less inclined to undertake more extensive lymphadenectomy in such cases. However, even cIA tumors may give lymph node metastases. Bott et al. reported an incidence of the N2 feature in patients with stage cI NSCLC of approximately 8.5%. Their study included patients operated from 1998 to 2010, in whom PET-CT examination was performed in only 12.5%. This was a limitation of the study that potentially contributed to a higher non-detection rate of nodal metastases preoperatively [35]. A recent study by Tsai et al. unveiled nodal upstaging in almost 8% of patients following VATS lobectomy with SMLND for cT1a-b NSCLC. Within this group, 82% of patients exhibited N2 metastases. All those patients had excellent preoperative nodal assessment, including PET-CT, and invasive mediastinal staging if mediastinal metastases were suspected [36]. These studies underscore the necessity for a comprehensive lymphadenectomy, even in patients with small nodules and good preoperative mediastinal staging. The significance of high-quality MLND and detection of nodal metastases is heightened in light of recent studies revealing promising results of adjuvant therapy after lung cancer surgery [37].

The surgeon’s experience greatly influences the results of minimally invasive lung cancer surgery. Numerous studies have demonstrated that with an increasing number of VATS lobectomies performed by the surgeon, there is a reduction in operating time, intraoperative blood loss, frequency of conversion to thoracotomy, and incidence of postoperative complications [38]. Ongoing discussions within the thoracic surgery community revolve around determining the number of operations required to overcome the learning curve, with estimates ranging from 25 to over 200 operations [39]. For the purposes of this study, we adopted a value of 50 surgical procedures, well established in the literature [40,41,42]. Most studies on VATS lobectomy learning curve focused on early postoperative outcomes. The relationship between the learning curve and lymphadenectomy accuracy has been infrequently explored. This study demonstrated that surgeons with more experience in VATS lobectomy were more likely to perform L-SMLND. Additionally, patients with L-SMLND had a greater number of lymph nodes and lymph node stations removed. These findings align with those of Mazzella et al., who reported that less experienced surgeons more frequently performed lymph node sampling instead of lymphadenectomy, harvested fewer lymph nodes, and performed fewer radical resections [43]. Lymph node dissection plays a crucial role in the surgical treatment of lung cancer as it improves staging accuracy and influences postoperative oncological treatment. During the learning phase of VATS lobectomy, attention should be directed not only to the anatomical resection itself, but also to the meticulous execution of lymph node dissection.

A concerning observation was the deterioration in the quality of lymphadenectomy over the study period. In the initial phase, spanning from 2007 to 2010, the proportion of patients with L-SMLND stood at 42.2%, whereas in the most recent period (from 2019 to 2022) it declined to 25.7%. Upon dividing the cohort into four equal-sized groups, it was revealed that the incidence of L-SMLND in the oldest group was 29.2%, while in the most recent group it was 23.3%. These findings markedly differed from those previously published in the literature. Osarogiagbon et al. employed the number of removed lymph nodes as a proxy for lymphadenectomy quality in lung cancer surgery and found that it gradually increased over the span of 12 years covered by their study [44]. Krantz et al. used the criterion of 15 or more lymph nodes as an indicator of good-quality lymphadenectomy, showing a rising trend of likelihood of patients having more than 15 nodes assessed from the year 2004 to 2013 [45]. The improvement in the quality of lymphadenectomy over time was observed not only in patients undergoing lung cancer surgery, but also in those treated for extrapulmonary cancer, such as rectal cancer [46]. The reasons for the decline in the quality of lymphadenectomy over the years observed in our study remain unclear. One possible contributing factor could be the increasing number of surgeons performing VATS lobectomy, necessitating the overcoming of the learning curve and resulting in a reduction in the number of operations per surgeon per year. Other potential factors include economic pressure to increase the number of daily operations, leading to attempts to shorten the operation time, or inaccuracies in reporting data on lymph nodes collection. Further research is essential to elucidate the underlying causes and enhance the quality of MLND.

Several other variables, such as BMI, smoking history, COPD, EBUS-TBNA, mediastinoscopy, side of surgery, and type of lobectomy were significant in univariate analysis. Some of these, as along with other factors, have been previously identified in other studies as being associated with the quality of intraoperative lymph node assessment [27,33,34,35,45]. However, these factors will not be discussed here, as they lost significance in the multivariate analysis.

In this study, we also demonstrated that more extensive lymphadenectomy was not associated with an increased risk of postoperative complications. Existing literature confirms that a more extensive lymph node dissection does not necessarily increase the risk of postoperative complications [47,48,49]. However, during lymph node dissection, damage may occur to crucial anatomical structures, such as the superior vena cava, esophagus, intermediate or main bronchi, recurrent laryngeal nerve, phrenic nerve, thoracic duct, or others. This may result in severe bleeding, esophageal fistula, bronchial fistula, vocal cord paralysis, diaphragmatic paralysis, or chylothorax. The incidence of these adverse events is very low, which may complicate statistical analyses and lead to drawing false conclusions based on obtained results [50]. These rare, severe complications should rather be assessed on a case-by-case basis. In-depth knowledge of anatomy, good surgical technique, careful use of surgical instruments, especially electrosurgical equipment, and appropriate exposure and identification of anatomical structures allow for the avoidance of intraoperative complications in most cases [51].

This study demonstrated, that the thoroughness of lymph node assessment was generally low. Improving the quality of lymphadenectomy during lung cancer surgery is crucial for accurate staging, planning of adjuvant treatment, and achieving the best possible long-term treatment results. There are several measures that can be implemented to improve the quality of lymphadenectomy for lung cancer surgery.

Firstly, the surgical team, including both the surgeon and the assistant, should be well trained and experienced in performing minimally invasive lung surgery. Surgeons should have in-depth knowledge of the anatomy of the mediastinum, which will enable the location of individual anatomical structures and lymph nodes and will facilitate complete and safe lymphadenectomy. Nowadays, thanks to the abundance of written materials, anatomical atlases and video materials, access to information on how to perform high-quality lymphadenectomy is virtually unlimited.

Secondly, surgeons should possess a very good understanding of the principles of oncological thoracic surgery and current guidelines for the treatment of lung cancer. Several lung cancer organizations, such as the International Association for the Study of Lung Cancer (IASLC) and the National Comprehensive Cancer Network (NCCN) provide recommendations on which nodal sites should be removed and what criteria should be met for complete resection [16,30,52]. Heightened awareness of recommended quality standards regarding the MLND has been shown to improve the quality of lymph node assessment [53].

Thirdly, the use of various intraoperative techniques may enhance the accuracy of lymphadenectomies. Osarogiagbon et al. developed a lymph node collection kit and demonstrated its effectiveness in improving nodal staging quality [54]. Subsequent studies have shown that the use of this kit enhances the thoroughness of lymphadenectomy and overall survival after lung cancer surgery [53,55]. Advanced electrosurgical devices may also facilitate tissue dissection and increase the thoroughness of lymphadenectomy [27,56,57].

Fourth, quality control of surgical procedures is essential. One of the methods for controlling quality involves the mandatory entry of surgical care data into dedicated databases. The Polish Lung Cancer Study Group Database contains detailed data on many aspects of lung cancer treatment, including number of lymph nodes removed for each nodal station. Data are reported separately by thoracic surgeons and pathologists, providing a comprehensive assessment of the quality of lymphadenectomy performed by surgeons in each department. Another example of quality control measures is the ESTS Quality Certification Program [58]. Data entered by individual centers into the ESTS Database are used to calculate a composite performance score. Sufficiently high score is one of the conditions for obtaining the prestigious ESTS Accreditation. One of the variables used to calculate the score is the percentage of patients operated on for primary neoplastic disease submitted to systematic lymph node dissection. This provides some degree of external control over the quality of lymphadenectomy at participating centers. The annual publication of data in the form of an ESTS Database Annual Report is an additional mode of assessing the activities of the departments [59]. However, there are many issues related to reporting of the surgical data, such as incorrect identification of the lymph node groups during removal, lymph node fragmentation, variations of nodal dissection techniques and extent, and difficulties in interpreting surgical reports by individuals entering data into databases [60,61]. Further efforts are needed to improve and standardize data collection and usage for the assessment of the quality of thoracic surgical procedures.

The introduction of direct quality control methods could address some of these problems. The assessment of thoracic surgery quality traditionally relies mainly on indirect methods, such as the analysis of early and long-term treatment outcomes and, less frequently, on the analysis of patients’ quality of life. Methods for assessing the quality of lymphadenectomy, such as determining the number of lymph nodes and nodal stations, or meeting appropriate guidelines, are also indirect and do not allow for the evaluation of the surgical procedure’s course. In a recent survey involving surgeons, urologists, gynecologists, and trainees, the majority of respondents agreed that the current methods of surgical reporting was insufficient to meet future quality requirements. The authors concluded that implementing intraoperative video or audio recording should be considered [62]. Recently, the Society of American Gastrointestinal and Endoscopic Surgeons (SAGES) issued recommendations on video data acquisition, storage, sharing, use, and governance [63]. Despite the widespread use of video materials in thoracic surgery, they have not yet been widely employed for quality control. These issues may be important and interesting for thoracic surgeons and should be the subject of further research.

### Strengths and Limitations

The main strength of the study was the inclusion of all patients who underwent VATS lobectomy in one large European country. This allowed us to obtain a very large amount of data for analysis. By covering a large period of time, from 2007 to 2022, we were able to analyze the impact of the learning curve on the accuracy of lymphadenectomy and changes in accuracy over time. A great advantage was the good quality of data, especially regarding the surgical procedure and histopathological examination results. The multicenter nature of the study made it possible to reflect the variability of surgical proceedings between various departments.

The study also had several limitations. Most of these limitations were due to the fact that this was a retrospective, multicenter cohort study. First, although included in one database, data originated from multiple departments. Differences in patient populations, treatment protocols, and data collection methods across departments might have introduced confounders that affected the study results. Secondly, although the definitions of variables entered into the database were unambiguous, we cannot exclude unforeseen differences between departments. Those could have led to inconsistencies in the data that influenced the results. Thirdly, selection bias is a common limitation of retrospective studies. Given the results obtained, in this study, it may have concerned qualification for surgery by different surgeons depending on clinical stage, including the results of PET-CT scans. Moreover, retrospective studies do not allow establishment of causation. For this reason, the results of the study enable us to draw conclusions solely on the relationship between the variables and the quality of MLND, and not on causation. Finally, the study was conducted on patients operated on within one type of healthcare system. Generalizing the study results to patients operated on in other countries should take into account differences healthcare settings.

The quality of histopathological and surgical data was good, but there were some missing data in other areas. The database contains information on the fact of performing PET-CT, but not on maximal standardized uptake value of fluorodeoxyglucose in lymph nodes and tumor. Analyzing the relationship between those variables and the accuracy of lymph node assessment would be interesting and should be considered in further trials. Data on body weight and height have been entered into the database since 2019. For this reason, information on BMI was available only in one quarter of patients, and only those operated on recently. Although BMI is a risk factor for postoperative complications, we could not include this variable in propensity score matching. Other data not included in the database were the type of VATS approach (uniportal vs. multiportal), information on conversion to thoracotomy, and on lymph node fragmentation. It would also be interesting to assess the relationship between the expression of molecular biomarkers, and long-term outcomes and quality of lymphadenectomy. However, these data were incomplete, their accuracy varied significantly between centers, and we did not include them in the statistical analysis. All this information could be useful for the purposes of the current research, and its absence should also be considered a limitation of the study.

## 5. Conclusions

This study demonstrated that a significant number of patients operated on for early-stage NSCLC did not undergo a sufficiently thorough mediastinal lymph nodes dissection. A detailed lymphadenectomy allows a more accurate assessment of the NSCLC stage, a precise determination of the long-term prognosis, and the planning of adjuvant treatment. Therefore, it is imperative to take measures to improve the quality of mediastinal lymph nodes dissection.

All identified risk factors associated with less accurate lymphadenectomy, such as smaller nodule diameter, lack of PET-CT, and lower surgeon’s experience, underscore the pivotal role of the surgeon in ensuring the quality of nodal assessment. There is concern about the declining quality of lymphadenectomy in recent years, particularly in the context of rapidly evolving adjuvant therapy.

Meanwhile, more extensive lymph node dissection does not appear to increase the incidence of complications. However, this should not diminish the surgeon’s responsibility to exercise caution during surgery. It is essential to recognize that very rare, but serious complications, such as damage to the recurrent laryngeal nerve, phrenic nerve, esophagus, bronchi, or vessels, may not be captured in statistical analysis.

To address these concerns, it is crucial to implement actions focused on enhancing the accuracy of surgery, including lymphadenectomy, while maintaining or reducing the incidence of postoperative complications. This may include initiatives such as education and quality control of surgical operations.

## Figures and Tables

**Figure 1 cancers-16-00346-f001:**
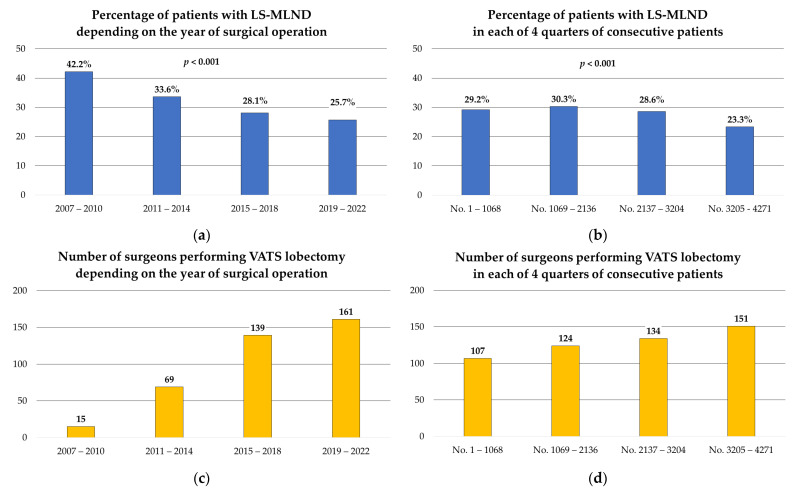
Changes in the accuracy of lymph nodes assessment and numbers of surgeons performing VATS lobectomy over time: (**a**) Percentage of patients with LS-MLND, depending on the year of surgical operation; (**b**) percentage of patients with LS-MLND in each of 4 quarters of consecutive patients; (**c**) number of surgeons performing VATS lobectomy, depending on the year of surgical operation; (**d**) number of surgeons performing VATS lobectomy in each of 4 quarters of consecutive patients.

**Table 1 cancers-16-00346-t001:** Univariable analysis of demographic and preoperative variables associated with compliance to the lobe-specific mediastinal lymphadenectomy guidelines.

	Lobe-Specific Lymph Node Dissection		*p*-Value
	Non L-SMLND (*n* = 3081)	L-SMLND (*n* = 1190)	
Age, years, mean (SD)	65.1 (SD: 8.1)	65.9 (SD: 7.8)	0.794
Male, *n* (%)	1408 (45.7)	568 (47.7)	0.233
Smoking history, *n* (%)	2052 (66.6)	934 (78.5)	<0.001 *
BMI, kg/m^2^, mean (SD)	26.9 (SD: 4.9)	27.6 (SD: 5.1)	0.040 *
ppFEV1%, mean (SD)	89.0 (SD: 19.4)	88.6 (SD: 19.1)	0.631
Comorbidities, *n* (%)	2444 (79.3)	932 (78.3)	0.469
Chronic obstructive pulmonary disease	542 (17.6)	274 (23.0)	<0.001 *
Coronary arterial disease	433 (14.1)	158 (13.3)	0.510
Cerebrovascular disease	40 (1.3)	23 (1.9)	0.123
Peripheral arterial disease	103 (3.3)	53 (4.5)	0.083
Hypertension	1571 (51.0)	605 (50.8)	0.930
Diabetes mellitus	462 (15.0)	185 (15.5)	0.652
Chronic kidney disease	24 (0.8)	16 (1.3)	0.085
Other neoplasm	502 (16.3)	173 (14.5)	0.159
ThRCRI, *n* (%)			0.806
Group A (*n* = 3622)	2610 (84.7)	1012 (85.1)	
Group B (*n* = 644)	469 (15.2)	175 (14.7)	
Group C (*n* = 5)	2 (0.1)	3 (0.2)	
CCI, median (IQR)	3 (IQR, 2 to 4)	3 (IQR, 2 to 4)	0.587
Patients with PET-CT	972 (31.5)	521 (43.8)	<0.001 *
Patients with EBUS-TBNA	364 (11.8)	114 (9.6)	0.038 *
Patients with mediastinoscopy	25 (0.8)	19 (1.6)	0.023 *

L-SMLND = lobe-specific mediastinal lymph node dissection; SD = standard deviation; BMI = body mass index; ppFEV1% = predicted postoperative percentage of calculated forced expiratory volume in 1 s; ThRCRI = Thoracic Revised Cardiac Risk Index; IQR = interquartile range; CCI = Charlson comorbidity index; PET-CT = positron emission tomography–computed tomography; EBUS-TBNA = endobronchial ultrasound-transbronchial needle aspiration. * Statistically significant (*p* < 0.05).

**Table 2 cancers-16-00346-t002:** Univariable analysis of surgery-related and histopathological variables associated with compliance to the lobe-specific mediastinal lymphadenectomy guidelines.

	Lobe-Specific Lymph Node Dissection		*p*-Value
	Non L-SMLND (*n* = 3081)	L-SMLND (*n* = 1190)	
Side of surgery, *n* (%)			<0.001 *
Right (*n* = 2543)	1719 (67.6)	824 (32.4)	
Left (*n* = 1728)	1362 (78.8)	366 (21.2)	
Type of surgery, *n* (%)			<0.001 *
Right upper lobectomy (*n* = 1415)	946 (66.9)	469 (33.1)	
Right middle lobectomy (*n* = 361)	266 (73.7)	95 (26.3)	
Right lower lobectomy (*n* = 739)	485 (65.6)	254 (34.4)	
Right upper bilobectomy (*n* = 14)	9 (64.3)	5 (35.7)	
Right lower bilobectomy (*n* = 14)	13 (92.9)	1 (7.1)	
Left upper lobectomy (*n* = 1055)	767 (72.7)	288 (27.3)	
Left lower lobectomy (*n* = 673)	595 (88.4)	78 (11.6)	
Surgeons’ experience, *n* (%)			0.003 *
Initial 50 VATS lobectomies (*n* = 2523)	1863 (73.8)	660 (26.2)	
Later lobectomies (>50) (*n* = 1748)	1218 (69.7)	530 (30.3)	
Histology, *n* (%)			0.059
Adenocarcinoma (*n* = 2363)	1694 (71.7)	669 (28.3)	
Squamous cell carcinoma (*n* = 1036)	731 (70.6)	305 (29.4)	
Other types (*n* = 872)	656 (75.2)	216 (24.8)	
pT, *n* (%)			<0.001 *
pT 1a (*n* = 442)	336 (76.0)	106 (24.0)	
pT 1b (*n* = 2136)	1566 (73.3)	570 (26.7)	
pT 1c (*n* = 1691)	1177 (69.6)	514 (30.4)	
Number of lymph nodes stations removed, median (IQR)			
N1 stations	2 (IQR, 1 to 2)	2 (IQR, 1 to 2)	<0.001 *
N2 stations	2 (IQR, 1 to 3)	4 (IQR, 3 to 4)	<0.001 *
Total number of lymph nodes stations	4 (IQR, 3 to 5)	6 (IQR, 5 to 6)	<0.001 *
Number of lymph nodes removed, median (IQR)			
N1 lymph nodes	4 (IQR, 2 to 6)	4 (IQR, 2 to 7)	<0.001 *
N2 lymph nodes	4 (IQR, 2 to 6)	8 (IQR, 5 to 12)	<0.001 *
Total number of lymph nodes	8 (IQR, 5 to 12)	12 (IQR, 8 to 18)	<0.001 *

L-SMLND = lobe-specific mediastinal lymph node dissection; IQR = interquartile range. * Statistically significant (*p* < 0.05).

**Table 3 cancers-16-00346-t003:** Results of the multivariate analysis of variables related to the L-SMLND for stage IA NSCLC.

	Odds Ratio	95% Confidence Interval	*p*-Value
Smoking	1.265	0.897 to 1.784	0.181
BMI	1.030	0.999 to 1.063	0.058
COPD	1.252	0.863 to 1.817	0.237
PET-CT	3.238	2.315 to 4.529	<0.001 *
Mediastinoscopy	1.934	0.119 to 31.561	0.643
EBUS-TBNA	0.889	0.572 to 1.380	0.599
pT	1.292	1.009 to 1.653	0.042 *
Surgeon’s experience	1.959	1.432 to 2.679	<0.001 *
No. of surgery (quarters)	0.647	0.474 to 0.884	0.006 *
Side of surgery	0.816	0.083 to 8.005	0.862
Right upper lobectomy	1.493	0.153 to 14.557	0.730
Right middle lobectomy	0.952	0.094 to 9.655	0.967
Right lower lobectomy	1.577	0.159 to 15.589	0.697
Right upper bilobectomy	3.570	0.128 to 99.477	0.453
Left upper lobectomy	1.306	0.133 to 12.856	0.819
Left lower lobectomy	0.422	0.041 to 4.333	0.468

BMI = body mass index; COPD = Chronic obstructive pulmonary disease; PET-CT = positron emission tomography–computed tomography; EBUS-TBNA = endobronchial ultrasound-transbronchial needle aspiration. * Statistically significant (*p* < 0.05).

**Table 4 cancers-16-00346-t004:** Propensity-matched analysis of non-L-SMLND versus L-SMLND: postoperative characteristics and complications data.

	Lobe-Specific Lymph Node Dissection		*p*-Value
	Non L-SMLND (*n* = 1184)	L-SMLND (*n* = 1184)	
Complications, *n* (%)	255 (21.5)	272 (23.5)	0.258
Prolonged air leak > 5 days	102 (8.6)	107 (9.0)	0.717
Residual air space	39 (3.3)	24 (2.0)	0.055
Re-drainage	35 (3.0)	25 (2.1)	0.191
Atrial arrythmia	39 (3.3)	31 (2.6)	0.332
Transfusion	37 (3.1)	43 (3.6)	0.495
Pneumonia	16 (1.4)	22 (1.9)	0.326
Bronchoscopy for atelectasis	7 (0.6)	14 (1.2)	0.125
Surgery for postoperative bleeding	9 (0.8)	12 (1.0)	0.511
Surgery for other postoperative complications	12 (1.0)	17 (1.4)	0.350
Delirium	5 (0.4)	6 (0.5)	0.762
Prolonged intubation	5 (0.4)	5 (0.4)	1.000
Bronchopleural fistula	1 (0.1)	1 (0.1)	1.000
Chylothorax	1 (0.1)	5 (0.4)	0.102
Recurrent laryngeal nerve palsy	1 (0.1)	2 (0.2)	0.563
Pulmonary embolism	1 (0.1)	4 (0.3)	0.179
Myocardial infarction	1 (0.1)	1 (0.1)	1.000
Cerebrovascular complications	1 (0.1)	4 (0.3)	0.179
Kidney failure	0	2 (0.2)	0.157
Other complications	22 (1.9)	25 (2.1)	0.658
Chest tube duration, days, median (IQR)	4 (IQR, 3 to 5)	3 (IQR, 2 to 5)	<0.001 *
Hospital stay, days, median (IQR)	6 (IQR, 4 to 7)	6 (IQR, 4 to 7)	0.870
In-hospital death	3 (0.3)	7 (0.6)	0.205

L-SMLND = lobe-specific mediastinal lymph node dissection; IQR = interquartile range. * Statistically significant (*p* < 0.05).

## Data Availability

The data presented in this study are available on request from the corresponding author.

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
