# Peer review of "Influencing Factors on the Quality of Lymph Node Dissection for Stage IA Non-Small Cell Lung Cancer: A Retrospective Nationwide Cohort Study"

_cancers, 2024, doi:10.3390/cancers16020346_

Round 1
Reviewer 1 Report
Comments and Suggestions for Authors
This retrospective, multicenter cohort study identified factors that influence the lymphadenectomy quality. The sample size is large, and the statistics process is good. Minor revisions are required. Here are some questions and suggestions.
1. In Figure 1, is it possible for you to add the relation between all the different treatments and the patients survival data?
2. In table 3, the authors listed most of the variables related to the L-SMLND for stage IA 241 NSCLC for multivariate analysis, I would like to ask is there any molecular biomarkers expression also working in the multivariate analysis?
Author Response
Dear Reviewer, we greatly appreciate your positive feedback on our manuscript. Thank you very much for all your comments. We responded to all comments and modified the manuscript accordingly. We hope that the changes we have made have had a positive impact on the quality of the manuscript.
Comment 1: In Figure 1, is it possible for you to add the relation between all the different treatments and the patients survival data?
Response and changes 1: In a previous study, the results of which were published in the journal "Cancers" this year (https://doi.org/10.3390/cancers15153877), we conducted a thorough analysis of the factors influencing the survival of patients after surgery for stage IA lung cancer. The study was based on the same database, the inclusion criterion was also VATS lobectomy in stage IA NSCLC, and the main difference was the range of dates covered by the study - in the previous study we aimed to assess the full five-year survival and the group was smaller, and in the current study we aimed to maximize the number of subjects to increase the power of the study. For this reason, we did not include survival analysis in the current study. To address this issue, we have modified a sentence in the introduction to stress the link between the accuracy of lymph node dissection and survival after early-stage lung cancer surgery: “More importantly, the study also found that L-SMLND was related to higher 5-year survival rates compared to patients with less comprehensive nodal assessment [25].”
Alternatively, we can perform survival analysis in L-SMLND and non-L-SMLND patient groups. However, this will replicate the results from the previous study, so we would prefer to avoid it.
Comment 2. In table 3, the authors listed most of the variables related to the L-SMLND for stage IA 241 NSCLC for multivariate analysis, I would like to ask is there any molecular biomarkers expression also working in the multivariate analysis?
Response and changes 2: Thank you for this comment. Molecular biomarker expression data were incomplete in the database, with the degree of accuracy varying greatly between departments. Therefore, we did not include these data in the statistical analysis. Nevertheless, the issue is undoubtedly interesting and should be the subject of further research.
We have added a sentence to the Limitations section: “It would also be interesting to assess the relationship between the expression of molecular biomarkers, and long-term outcomes and quality of lymphadenectomy. However, these data were incomplete, their accuracy varied significantly between centers, and we did not include them in the statistical analysis.”
Reviewer 2 Report
Comments and Suggestions for Authors
A rather thoroughly prepared manuscript regarding the quality of lymph node dissection in 4,271 patients who underwent minimally invasive surgical operations for early-stage lung cancer. Data comes from Poland.
Below are my comments:
1. I have no critical comments about the abstract and introduction, these subsections are well-written,
2. materials and methods:
a. please provide the decision number of the bioethics commission,
b. subchapter 2.1 - references are missing here, please complete them,
3. I have no major comments on the results, they are carefully prepared, and, perfectly, the propensity score was used.
4. the discussion is well written.
5. however, the entire manuscript must be edited by a native speaker.
6. references need to be expanded:
https://www.mdpi.com/2072-6694/15/14/3735
https://www.mdpi.com/2072-6694/15/13/3447
Comments on the Quality of English LanguageModerate editing of the English language is required.
Author Response
Dear Reviewer, we greatly appreciate your positive feedback on our manuscript. Thank you very much for all your comments. We responded to all comments and modified the manuscript accordingly. We hope that the changes we have made have had a positive impact on the quality of the manuscript.
Comment 1: I have no critical comments about the abstract and introduction, these subsections are well-written.
Response and changes 1: Thank you very much for this comment.
Comment 2a. materials and methods: please provide the decision number of the bioethics commission.
Response and changes 2a: According to the guidelines of the Bioethics Committee of the Poznań University of Medical Sciences, most retrospective studies are not medical experiments. In such cases, the commission issues a written opinion that the study does not bear the characteristics of a medical experiment and that there is no need to obtain consent for the study, and the Committee does not assign such a decision a specific number. For our study, the decision of the Bioethics Committee was issued on June 14, 2023. We have attached the committee's decision as a supplementary material and noted this fact in the Material and Methods section.
Comment 2b. materials and methods: subchapter 2.1 - references are missing here, please complete them.
Response and changes 2b: We have added a reference to the most recent report by the Polish National Cancer Registry, which is the official national organization that gathers all information on diagnosis and treatment of lung cancer in Poland.
Comment 3. I have no major comments on the results, they are carefully prepared, and, perfectly, the propensity score was used.
Response 3: Thank you very much for this comment.
Changes 3: No changes required.
Comment 4. the discussion is well written.
Response 4: Thank you very much for this comment.
Changes 4: No changes required.
Comment 5. however, the entire manuscript must be edited by a native speaker.
Response and changes 5: Thank you very much for this comment. We have the manuscript edited by a native speaker. Many changes have been introduced and we have included them in our text.
Comment 6. references need to be expanded:
https://www.mdpi.com/2072-6694/15/14/3735
https://www.mdpi.com/2072-6694/15/13/3447
Response and changes 6: Thank you for the comment, both references are relevant for the study, and we have included these in the manuscript.
We included the reference https://www.mdpi.com/2072-6694/15/14/3735 to support the statement: “There is no consensus if one type of MLND is superior to the other [17].”
We have added a sentence with reference https://www.mdpi.com/2072-6694/15/13/3447 to the discussion: “An increase in the quality of lymphadenectomy over time was observed not only in patients operated on for lung cancer, but also for extrapulmonary cancers, such as rectal cancer [45].”
Round 2
Reviewer 2 Report
Comments and Suggestions for Authors
The authors have addressed all my comments for this paper. Congratulations!